# Impact of Infections During Pregnancy on Transplacental Antibody Transfer

**DOI:** 10.3390/vaccines12101199

**Published:** 2024-10-21

**Authors:** Celeste Coler, Elana King-Nakaoka, Emma Every, Sophia Chima, Ashley Vong, Briana Del Rosario, Roslyn VanAbel, Kristina M. Adams Waldorf

**Affiliations:** 1School of Medicine, University of Washington, Seattle, WA 98195, USA; ccoler23@uw.edu (C.C.); elanakn@uw.edu (E.K.-N.); eevery@uw.edu (E.E.); 2Department of Obstetrics and Gynecology, University of Washington, Seattle, WA 98109, USA; schima3@uw.edu (S.C.); axvong@outlook.com (A.V.); brianadr@uw.edu (B.D.R.); 3Department of Global Health, University of Washington, Seattle, WA 98105, USA; 4College of Pharmacy, University of Michigan, Ann Arbor, MI 48109, USA; roslyn.a.vanabel@gmail.com

**Keywords:** pregnancy, fetus, antibody, placenta, antibody, vaccine, SARS-CoV-2, malaria, HIV

## Abstract

Vaccination in pregnancy is important to protect the mother and fetus from infectious diseases. The transfer of maternal antibodies across the placenta during pregnancy can continue to protect the neonate for several months after birth while the neonatal adaptive immune system develops. Several pathogens have been shown to impair the transplacental transfer of maternal antibodies, including human immunodeficiency virus, malaria, the severe acute respiratory syndrome coronavirus 2, and cytomegalovirus. This review discusses the mechanisms contributing to decreased transplacental antibody transfer in the setting of maternal infections, such as changes in antibody glycosylation profile, maternal hypergammaglobulinemia, and placental injury. The frequency of epidemics is increasing, and pregnant people are more likely to become exposed to novel pathogens now than they were in the past. Understanding the mechanisms by which infectious diseases impair maternal–fetal antibody transfer is important for pandemic preparedness to maximize the impact of maternal vaccination for child health.

## 1. Introduction

Vaccination during pregnancy provides immune protection for both the mother and fetus through the transfer of maternal antibodies across the placenta. The passive transfer of maternal antibodies to the fetus during pregnancy is particularly important to protect the neonate from infectious diseases for several months after birth, until maternal antibodies decay. This is critical because the neonate is highly vulnerable to infectious diseases during the first months of life due to an immature adaptive immune system in neonates and infants [1] and blunting of vaccine responses by maternal antibodies [2,3,4,5]. Societies representing obstetricians and pediatricians recommend vaccination of pregnant mothers with several vaccines [e.g., tetanus, diphtheria, and acellular pertussis (Tdap), seasonal influenza vaccine, and severe acute respiratory syndrome coronavirus-2 (SARS-CoV-2), Table 1] [6,7,8,9,10,11]. Recently, the respiratory syncytial virus vaccine (RSV) has also been recommended in some pregnancies, depending on whether the due date falls within the RSV season [12]. Maximizing the impact of maternal vaccination for the benefit of the neonate requires knowledge of the factors that enhance and impair the placental transfer of antibodies.

Infectious diseases can impair the transplacental transfer of maternal antibodies through changes to the glycosylation profile of the maternal antibody, placental injury, and the induction of maternal hypergammaglobulinemia. Although the sequelae of maternal infections on placental antibody transfer are known for a few pathogens, they are poorly understood. New epidemics and pandemics are occurring with increasing frequency due to airplane travel, environmental change that increases human interaction with animal reservoirs, and a greater population density in overcrowded and unhygienic environments [13]. Globally, pregnant people are more likely to become exposed to novel pathogens that may impair transplacental antibody transfer than they were in the past.

Pandemic preparedness for pregnant people requires a review of what is currently known regarding the impact of infectious diseases on transplacental antibody transfer and the mechanism by which impairment occurs. The objective of this review is to provide information that can help guide potential changes in maternal vaccination during an epidemic, as well as a broader research agenda. Herein, we focus on what is known regarding transplacental antibody transfer in normal pregnancy, key pathogens that impair antibody transfer, mechanisms by which the impairment occurs, and critical knowledge gaps in the field. Although passive immunity through maternal breastmilk contributes to neonatal immunity, it is unknown to the obstetrician whether a mother will or will not exclusively breastfeed and the duration of breastfeeding. Therefore, we have focused this review on transplacental antibody transfer during pregnancy and the negative impact of infectious disease.

## 2. Normal Transplacental Antibody Transfer

Achieving immune homeostasis and protection from pathogens requires coordinated actions of the innate and adaptive immune system. However, in neonates, mechanisms of adaptive immunity are limited, and newborns rely heavily on innate immune defenses [14]. The transplacental transfer of maternal IgG antibodies to the fetus is a key component of the neonate’s first line of immune defense. Unfortunately, the placenta lacks specific transporters for IgA, preventing its transfer to the fetus. However, IgA is the predominant immunoglobulin in breast milk, which is important in shaping the infant’s gut microbiota and protecting against pathogens after birth [15].

Transplacental antibody transfer occurs primarily through the interaction of the Fc domain of maternal IgG antibodies and the neonatal Fc-receptor (FcRn) expressed on placental syncytiotrophoblast cells [16,17,18]. FcRn consists of a beta-2 microglobulin subunit and a heavy chain with three extracellular domains (α1, α 2, α 3) that form a binding site for the IgG Fc region [19,20]. FcRn binds to IgG in a pH-dependent manner, with a high affinity for IgG at pH 6.0–6.5 [21,22]. Once IgG binds to FcRn, IgG is taken up from the maternal circulation by syncytiotrophoblast cells and packaged into acidic endosomes [23]. Bound IgG is then transcytosed to the basolateral, fetal side of the syncytiotrophoblast and released upon exposure to a physiological pH (7.4; Figure 1) [24]. After the maternal IgG is released, the FcRn molecule is recycled back to the syncytiotrophoblast membrane [25]. This process relies upon the pH-depending binding of IgG to FcRn to transport IgG down a concentration gradient and through a cell layer [24,26,27].

The expression of FcRn receptors is determined by gestational age, first appearing early in fetal development before increasing progressively from early in the second trimester until the end of pregnancy [28]. Similarly, the transfer of IgG from mother to fetus begins early in the second trimester and increases as the pregnancy progresses, with the greatest transfer in the third trimester [28]. Accordingly, there is a reduced transfer of IgG in preterm compared to full-term neonates [29,30]. In a study of 107 paired maternal and fetal blood samples, IgG levels were shown to rise continuously between 17 and 41 weeks’ gestation [31]. In the study, fetal IgG increased from 5–10% of maternal levels between 17 and 22 weeks to 50% of maternal levels by weeks 28–32. The majority of IgG is transferred in the final four weeks of pregnancy, with fetal IgG concentrations exceeding maternal concentrations by nearly 30% at full term. Multiple studies have corroborated these findings using cord/maternal antibody ratios as a measurement of antibody transfer efficiency. These studies have found that at 33–36 weeks’ gestation, maternal/cord antibody ratios are approximately 1.0, indicating an equal concentration of antibodies in the mother and fetus [15,31]. By 37–41 weeks’ gestation, the maternal/cord antibody ratio is, on average, 1.5, reflecting a greater concentration of maternal antibodies in the fetus than in the mother. Accumulation of maternal antibodies in the fetus over time is an important reason why vaccination in pregnancy is recommended for TdAP as early in the 27–36 weeks of gestation window as possible [7].

Among the five antibody classes (IgG, IgM, IgA, IgD, IgE), IgG is the only antibody class that is transferred across the placenta in significant quantities and is prevalent in neonatal serum and mucosal tissues such as the airways, distal gastrointestinal tract, and genitourinary tract [32]. IgG is further divided into four subclasses, IgG1, IgG2, IgG3, and IgG4. Subclasses also vary in their serum half-life; the half-life of IgG1, IgG2, and IgG4 is, on average, 23 days compared to IgG3, which has a shorter half-life of 2–6 days [17,33]. Each subclass also crosses the placenta with varying efficiency. Despite multiple studies researching the efficiency of subclass transplacental transfer in uninfected pregnant people, there is no consensus on which subclass predominates (Table 2) [34]. Most studies have found the greatest transfer efficiency was for the IgG1 subclass and the lowest transfer efficiency was for IgG2 [34]. Varying reports of transfer efficiency may be due to differences in placental weight, birth weight, gestational age, and maternal antigen exposure between study populations. Understanding the variation among subclasses is important, as each subclass responds differently to pathogens and vaccines [35,36].

The same mechanism that transfers protective IgG across the placenta also allows natural and pathological maternal antibodies to enter the fetal circulation. Maternal autoantibodies can also transfer to the fetus and may increase the odds for the child to develop autism spectrum disorder, brain injury, or an autoimmune disease later in life [45,46,47]. In Rh alloimmunization, the fetus can be killed by maternal antibodies that target paternally inherited minor blood group antigens [48]. Additionally, neonatal Graves’ disease is caused by thyroid-stimulating receptor antibodies that cross the placenta and target the fetal thyroid gland [49]. Alternatively, the fetus and neonate may manifest a disease (i.e., congenital heart block with lupus) that typically resolves over several months as the maternal antibody concentrations decline [50]. Thus, transplacental antibody transfer has implications for the neonate that extend beyond the provision of anti-infectious protection.

## 3. Pathogens That Impair Transplacental Antibody Transfer

### 3.1. HIV/AIDS

Globally, there were 1.2 million pregnant women with HIV in 2023, of which an estimated 84% received antiretroviral therapy (ART) [51]. The administration of ART has led to a significant decrease in vertical HIV transmission worldwide. Infants born to HIV-infected women are HIV exposed, but the majority remain uninfected [52]. Still, there is compelling evidence that HIV-exposed uninfected infants are at increased risk for vaccine-preventable diseases compared to their HIV-unexposed counterparts in the first 6 months of life, including viral and invasive bacterial infections, such as Streptococcus pneumoniae and Group B Streptococcus [53,54].

Interpreting data that compares HIV-exposed uninfected infants to HIV-unexposed infants is challenging due to variable access to ART and disparate breastfeeding rates between groups. Meta-analyses from sub-Saharan Africa have found an increased risk of pneumonia, diarrheal illness, and death in HIV-exposed uninfected infants compared to HIV-unexposed infants [55,56]. One meta-analysis of 22 mostly sub-Saharan African cohorts with more than 29,000 children found a 60–70% increased risk of death for up to five years of age in HIV-exposed uninfected children compared to HIV-unexposed children [55]. Studies conducted after 2002, when ART was more widely available, continued to show a 50% increase in mortality in the HIV-exposed uninfected cohort (RR 1.46, 95% CI 1.14–1.87) [55].

The reason for an increased infection risk for HIV-exposed uninfected infants is unknown but may be a function of exposure to ART, a maternal immune environment shaped by HIV infection and treatment, the frequency of maternal vaccination, breastfeeding rates, placental insufficiency, or reduced transplacental antibody transfer. Although the answer is likely multifactorial, many studies indicate that maternal HIV infection reduces transplacental antibody transfer [57,58,59,60,61,62]. A study of 109 maternal–infant pairs evaluating cord/maternal antibody ratios found that placental transfer of maternal antibodies was reduced by 40% for pertussis-specific, 23% for Hemophilus influenza type b (Hib)-specific, and 27% for tetanus-specific antibodies versus HIV-uninfected mothers [62]. The antibody transfer process is impaired on multiple fronts, beginning with lower specific antibody titers in pregnant women with HIV due to HIV-induced alterations in maternal B cell activity [57,63].

B cell modulation in response to HIV is complex, and B cell activity varies based on the timing, duration, and treatment status of an HIV infection. For example, in early HIV infection, plasmablasts are rapidly produced and act as short-lived effector cells of an early antibody response [64]. In chronic HIV infection, there is a greater proportion of immature and exhausted B cells, which have a reduced capacity to make antibodies [65]. Overall, in an untreated HIV infection, peripheral blood B cell counts are lower in HIV-infected individuals compared to their uninfected counterparts [66]. With lower B cell counts, the ability to produce antibodies in response to vaccination or infection is impaired, resulting in a lesser IgG response to pathogens. After ART, B cells have a better functional profile, with B cell numbers increasing in tandem with a reduction in HIV viral load. ART also helps normalize B cell subpopulations, resulting in greater antibody production, which ultimately benefits both mother and fetus [66]. As such, the length of time a person has been on ART is likely another factor determining how functional their B cells are. These considerations complicate cohort studies of maternal–fetal antibody transfer in pregnant women living with HIV who may have a combination of untreated and treated HIV in different stages.

In addition to HIV-positive mothers having lower antibody titers at baseline, the transfer of these antibodies across the placenta is impaired, resulting in overall lower mean cord/maternal antibody titer ratios and higher numbers of infants born with antibodies below protective levels [63]. While the exact mechanism of this impaired transport is unknown, hypergammaglobulinemia, antibody glycosylation, and placental inflammation in pregnant women living with HIV may play a role [63].

### 3.2. Malaria

According to the 2023 World Malaria Report, 36% of pregnancies were infected with malaria within 33 moderate- and high-transmission countries in the World Health Organization (WHO) African Region; this represents an estimated 12.7 million malaria infections during pregnancy [67]. Malarial infection in pregnancy within these 33 countries is estimated to have resulted in 914,000 neonates of low birth weight, a risk factor for neonatal mortality. The primary mechanism of neonatal morbidity and mortality is thought to be caused by direct placental infection, referred to as placental malaria, as confirmed by histopathologic evidence of parasitized erythrocytes in placental tissue [68].

In addition to causing placental inflammation and dysregulated placental development, there is strong evidence that placental malaria contributes to poor outcomes in the neonate by reducing antibody transfer [43,63,69,70,71,72]. In a case–control study comparing maternal antibody concentrations in maternal and cord blood of mothers with and without placenta malaria, there was a significant reduction of 17–31% in the placental transfer of maternal antibodies to five P. falciparum recombinant antigens [73]. In addition to a reduction in the transfer of malaria-specific IgG, one study has shown that placental malaria is associated with a reduction in transplacental antibody transfer of antibodies to herpesvirus type 1, RSV, and varicella zoster virus of 69%, 58%, and 55%, respectively [43].

Many of the studies assessing malaria’s impact on transplacental antibody transfer are studies of pregnant women with an HIV co-infection. There is considerable evidence that HIV and malaria co-infections act synergistically; individuals with HIV are more susceptible to symptomatic malaria, and those with malaria experience higher viral loads and lower CD4+ counts [74,75]. Importantly, many studies have found that HIV infection is associated with reduced levels of cord blood antibodies, particularly IgG1, against multiple antimalarial antigens due to altered antibody levels in the mother and diminished transplacental transfer of antibodies [76,77]. Taken together, these studies suggest that in areas of high HIV seroprevalence, both maternal and infant health may be severely affected by malaria. This further validates our need for a better understanding of maternofetal transfer to design better preventative measures and treatment strategies that benefit maternal and child health.

### 3.3. COVID-19

Maternal–fetal transmission of SARS-CoV-2 is thought to occur in approximately 2% of infected pregnant women [78]. More often, early neonatal transmission from an infected mother is observed [79]. Fortunately, severe SARS-CoV-2 infection is uncommon in infants and neonates [80,81,82]. Still, pediatric populations are vulnerable to poor outcomes following SARS-CoV-2 infection, such as neonatal death, extended perinatal mortality, and low Apgar score [83].

Either vaccination or natural infection can result in the transfer of SARS-CoV-2 antibodies across the placenta. A study of 582 maternal–newborn dyads found that maternal vaccination against SARS-CoV-2 was associated with higher concentrations of IgG antibodies to the SARS-CoV-2 spike protein in both maternal and cord blood compared to a symptomatic or asymptomatic natural SARS-CoV-2 infection [84]. Other studies in the adult population support that vaccination produces higher concentrations of antibodies than natural infection [85,86]. This further demonstrates the protective benefit of vaccination for the mother and neonate.

There is evidence to suggest that maternal SARS-CoV-2 infection can also compromise the transfer of antibodies to the fetus. A study utilizing systems serology to analyze the Fc profile of antibodies specific to influenza, pertussis, and SARS-CoV-2 found that SARS-CoV-2-specific antibodies were transferred across than placenta in significantly lower quantities than influenza and pertussis antibodies. Additionally, the cord titers and functional activity of these antibodies were lower than those present in maternal plasma [87]. These observations were exclusive to the third trimester, highlighting a vulnerability to third-trimester infection.

Interestingly, the effects of maternal SARS-CoV-2 infection on the fetus seem to be sex specific. Maternal SARS-CoV-2 infection has been associated with a higher risk for stillbirth and early neonatal mortality in male fetuses, particularly in the first week of life [88,89,90]. Studies have speculated that this pattern to be attributed to vulnerability to chromosomal abnormalities, genes on the Y chromosome, and the faster development and high metabolic rates of male embryos [91,92]. Studies highlighting the link between maternal SARS-CoV-2 infection and reduced placental antibody transfer in male pregnancies provide an alternative explanation for the sex-specific differences observed in SARS-CoV-2 disease severity. In a study involving 68 pregnant participants with equal numbers of male and female fetuses, maternal IgG titers against SARS-CoV-2-specific antigens were significantly lower in pregnancies with male versus female fetuses [93]. Although greater placental FcRn expression was observed in male pregnancies, which should improve transplacental IgG transfer, cord/maternal antibody ratios for SARS-CoV-2 were lower in male versus female fetuses. Interestingly, these male fetuses had higher titers of antibodies modified by fucosylation and galactosylation, which are less efficiently transferred by FcRn. The role of fucosylation and galactosylation in antibody transfer is discussed in greater detail in the next section of this review.

Evidence of sexually dimorphic transfer of maternal humoral immunity in maternal SARS-CoV2 infection may provide an explanation for the increased vulnerability of male neonates to SARS-CoV-2 and have sex-specific implications for SARS-CoV-2 vaccination. Future studies may elucidate these findings across gestation and explain their long-term implications.

## 4. Infection-Associated Mechanisms Impairing Maternal–Fetal Antibody Transfer

The causal mechanisms behind reduced antibody transfer efficiency are an active and growing area of research. Pathogens can utilize many mechanisms to disrupt the transfer of antibodies from mother to fetus, which is crucial for providing newborns with passive immunity (Figure 2). The variability and overlap of these mechanisms, which include hypergammaglobulinemia, glycosylation, placental inflammation and injury, and impaired trophoblast function, highlight the complexity of maternal–fetal antibody transfer and the value of studying how infections can disrupt this critical process.

### 4.1. Hypergammaglobulinemia

Several pathogens are known to be associated with hypergammaglobulinemia, wherein pregnant individuals develop high levels of non-specific and occasionally functionally impaired IgG that saturate the finite Fc receptors [15,30,43,69,70,94]. Various thresholds have been used to define hypergammaglobulinemia, and concentrations of IgG > 1500, >1600, or 1700 mg/dL are the most common [15,69,94,95]. While hypergammaglobulinemia has been recognized as an independent mechanism of impaired transplacental antibody transfer for some time, several studies in recent years now indicate that HIV and malaria infection during pregnancy may induce hypergammaglobulinemia and reduce transplacental antibody transfer [69,95,96,97]. While studies have shown that hypergammaglobulinemia is associated with a more severe disease course [98], its role in reduced transplacental antibody transfer has not yet been shown in SARS-CoV-2-infected pregnant women.

In HIV and placental malarial infection, hypergammaglobulinemia is induced via abnormal B cell activation and IgG overproduction [99,100]. In the setting of placental malaria, compelling evidence for the role of hypergammaglobulinemia in reduced antibody transfer includes a study of 325 mother–infant pairs in Papua New Guinea. The study found that placental malaria was associated with increased odds of cord antibodies in the lowest tercile, and this association was compounded by maternal titer and hypergammaglobulinemia. Hypergammaglobulinemia was strongly associated with increased odds of low cord antibodies in both univariable and multivariable models [adjusted OR 5.09 (95% Cl: 1.95–13.32), *p* < 0.001]. This suggests that the induction of hypergammaglobulinemia by placental malaria is a major driver of reduced antibody transfer.

The induction of hypergammaglobulinemia by maternal HIV infection has also been shown to contribute to reduced antibody transfer. In a case–controlled study of maternal and cord total IgG, IgG subclass, and cord-to-maternal ratios in 103 HIV-infected pregnant women from Cameroon, HIV-associated hypergammaglobulinemia was shown to significantly reduce the transplacental transfer of malaria-specific antibodies [97]. The study found that HIV-infected mothers who developed hypergammaglobulinemia had a greater reduction in cord blood to maternal blood IgG ratios compared to HIV-infected mothers without hypergammaglobulinemia. One challenge in studying hypergammaglobulinemia in HIV and malaria is parsing out the independent effects of HIV infection, malaria infection, and hypergammaglobulinemia due to the overlapping prevalence of these conditions in study populations [63]. By studying antibody transfer in malaria-uninfected and HIV-infected mothers with and without concurrent hypergammaglobulinemia, this study fulfills a key knowledge gap.

### 4.2. Antibody Glycosylation

Another factor influencing antibody transfer efficiency is the glycosylation of the antibody Fc region [101]. Antibody glycosylation, a post-translational modification, enables the selective transfer of specific antibody subpopulations. In healthy pregnancies, galactosylation and sialyation levels increase and fucosylation levels decrease in the IgG Fc region [93]. In neonates, studies have shown a decrease in the concentration of non-glycosylated IgG and an increase in the concentration of galactosylated IgG, which suggests that glycosylated antibodies are transported preferentially [36].

Early in the SARS-CoV-2 pandemic, a study showed that SARS-CoV-2 induces alterations of Fc-glycan profiles [87]. This prompted a study of how differences in Fc-glycan profiles in the setting of maternal SARS-CoV-2 influence transplacental antibody transfer. Perturbations in Fc glycosylation were observed when the SARS-CoV-2 infection occurred in the third, but not second trimester, suggesting that antibody transfer might normalize over time [87]. They found enhanced digalactyslation and fucosylation and reduced agalatosylation on S-specific antibodies. While alterations of Fc-glycan profiles typically offer disease-specific antibodies the ability to recruit innate immune effector functions aimed at controlling pathogens more effectively [102,103], these alterations resulted in reduced transfer. Simultaneously, there were infection-induced increases in IgG and increased expression of FCGR3A on the placenta. While these compensatory mechanisms did not offset the effects of antibody glycosylation, this may provide interesting insights into maternal vaccine design.

Interestingly, antibody glycosylation in maternal SARS-CoV-2 infection may explain sex-specific differences in transplacental antibody transfer and neonatal disease severity [93]. In a study of maternal and cord blood as well as fetal-side placental biopsies, placentas of SARS-CoV-2-positive male pregnancies showed an increased expression of the Fc-y receptor III (FCyRIII). Fucosylated antibodies, such as the S-specific antibodies in SARS-CoV-2-infected mothers as previously described, have been shown to transfer less efficiently through the FCyRIII [104,105,106]. As a result, males primarily transferred afucosylated antibodies, which were relatively scarce among the spike protein-specific antibodies. Pregnancies with female fetuses, in contrast, showed higher maternal levels of SARS-CoV-2-specific antibodies and reduced FCγRIII placental expression. This sex-specific difference facilitated a more efficient transfer of antibodies through the placenta based on the available Fc-glycan profile of spike protein-specific antibodies.

Similar observations have been made in the context of maternal HIV infection [107,108]. In a study of 48 maternal/neonate dyads, elevated levels of agalactosylated antibodies and reduced levels of digalactosylated and sialylated antibodies were observed in pregnant women with HIV, resulting in a commensurate reduction in transplacental transfer [108]. Interestingly, in a study of distinct HIV vaccine regimens, immunization was shown to generate antigen-specific antibodies with comparable antibody glycosylation patterns, induce different antigen-specific IgG glycosylation profiles, and overcome differences in bulk IgG glycosylation [103]. This indicates that distinct inflammatory signals during B cell priming can influence antibody glycosylation, offering a promising means by which vaccine regimens can enhance the antiviral activity of the innate immune system to protect the mother and fetus during infection.

### 4.3. Placental Inflammation and Impaired Trophoblast Function

An intact functioning placenta is essential for a healthy pregnancy, as it forms the primary interface between mother and fetus. Studies indicate that a wide spectrum of viruses can injure the placenta and impact placental nutrient and oxygen transport, including Zika virus (ZIKV) [109,110,111], dengue virus [112], cytomegalovirus (CMV) [113,114], SARS-CoV-2 [115,116,117], and malaria [118,119,120,121]. The extent to which many viruses impact the placental transfer of maternal antibodies and how trophoblast injury might impact antibody transfer is poorly understood.

Although CMV is typically not pathogenic in immunocompetent people, an infection in pregnant individuals can cause placental damage as well as fetal injury or death [122]. CMV is known to transcytose across syncytiotrophoblasts in low-avidity IgG-virion complexes, which then promote infection of underlying cytotrophoblasts. From there, infection spreads to susceptible stromal fibroblasts and endothelial cells in the villous core before crossing into the fetal circulation [123]. There are two mechanisms by which CMV has been shown to cause placental injury. First, CMV infection can dysregulate proteins critical for self-renewal and differentiation of trophoblast progenitor stem cells, leading to impaired placental development. Second, CMV infection can cause trophoblast apoptosis through tumor necrosis factor secretion, ultimately causing placental villitis [113,124,125,126]. This can lead to various forms of injury, including placental edema, fibrosis, thickening and abruption, intrauterine growth restriction, and pre-eclampsia [127]. In a study of 93 mother–infant pairs recruited from 2006 to 2018, maternal CMV infection was associated with a 23% decrease in antibody transfer efficiency. A mediation analysis determined that 74.5% of this reduction was due to maternal hypergammaglobulinemia. The study speculated that CMV might modify antibody transfer efficiency through additional mechanisms, including CMV-induced placental inflammation and syncytiotrophoblast damage, soliciting future studies to explore whether placental inflammation disrupts efficient IgG shuttling across the placenta [128].

Several studies have shown that SARS-CoV-2 infection during pregnancy can cause “SARS-CoV-2 placentitis” characterized by placental inflammation and injury. Common characteristics of SARS-CoV-2 placentitis include perivillous fibrin deposition, vascular malperfusion, and trophoblastic necrosis [129,130]. While it has been established that SARS-CoV-2-induced placental damage contributes to an increased risk of fetal morbidity and mortality [131,132], whether damage to the placenta contributes to the reduced antibody transfer efficiency observed with SARS-CoV-2 infection is unknown.

There are few studies on the effect of placental inflammation and transplacental antibody transfer in the settling of maternal placental malaria infection. It is possible that malaria can adversely impact placental antibody transfer by inducing placental inflammation, depositing malaria pigment within inflammatory cells, and thickening the trophoblast basement membrane. In a study of placental malaria where mother–cord paired serum samples were collected at delivery, the detection of monocytes containing malaria pigment showed a greater reduction in antibody transfer efficiency compared to placental infections that did not contain monocytes [133]. This suggests that placental inflammation is likely involved in reduced maternofetal transfer. There is even less data on the effect of HIV infection-induced acute and chronic placental inflammation and its contribution to impaired antibody transfer, highlighting an important knowledge gap [134,135].

## 5. Knowledge Gaps

Despite many studies identifying the biological factors and pathogens that influence transplacental antibody transfer, key knowledge gaps remain. The role of hypergammaglobulinemia in impaired antibody transfer represents a critical gap in our knowledge. Although this review focuses on hypergammaglobulinemia in the context of HIV and malaria, hypergammaglobulinemia has been observed in the setting of other viral infections, liver disease, hematologic disorders, and autoimmune conditions [136,137,138]. Future studies should further characterize the role of hypergammaglobulinemia in reduced transplacental antibody transfer both independently and in the setting of other conditions associated with reduced transfer (i.e., maternal infection, inflammation, preterm birth). The evidence is also inconsistent on whether treating the cause of hypergammaglobulinemia can mitigate its effects on transplacental antibody transfer [54,139,140,141,142]. For example, in a study performed on HIV-positive pregnant women in Malawi, 24 months of ART was associated with reduced total IgG over time; however, it did not reduce total IgG below the threshold of hypergammaglobulinemia [140]. Shorter courses of ART (7–10 months) yielded a similar reduction in IgG, but this reduction was not maintained over time. Understanding how the treatment of hypergammaglobulinemia can mitigate its effects on reduced antibody transfer can inform prevention and intervention strategies. Addressing this knowledge gap becomes crucial to evaluate the efficacy of prophylactics and therapeutics to prevent impairment of antibody transfer after infection with emerging viruses (Mpox, Oropouche virus). Finally, there are several studies exploring the impact of viral infections and pregnancy-related disorders, such as pre-eclampsia, on placental health; however, their influence on transplacental antibody transfer efficiency is relatively unknown and requires further investigation [129,143,144].

While there are few studies examining the influence of maternal antiviral responses on antibody transfer, one study found intriguing results on the impact of maternal inflammation on antibody transfer [145]. In this study, nasopharyngeal swabs from newborns, cord blood, and maternal blood samples were collected on the day of delivery from fifty SARS-CoV-2-positive pregnant patients, nine SARS-CoV-2 mRNA-vaccinated pregnant patients, and twelve noninfected pregnant controls. The transfer of anti-RBD IgG was found to be significantly inversely correlated with IL-6 but positively correlated to IL-10 and IL-23. These findings necessitate more information on the degree to which antibody transfer is correlated with inflammation and how cytokines influence the transfer of protective antibodies.

Finally, future studies are needed to elucidate how maternal IgG levels, IgG subclass, and antibody transfer efficiency impact the duration of protection in newborns. This will require detailed vaccine studies in pregnancy and will prepare the foundation for understanding how antibody profiles and glycosylation patterns translate to protection for the neonate after birth. Together, these proposed research efforts may provide more clarity on neonate protection against multiple pathogens in the coming years.

## 6. Conclusions

There is strong evidence that infection with SARS-CoV-2, malaria, HIV, and CMV during pregnancy may reduce the transfer of maternal antibodies to the fetus through hypergammaglobulinemia, changes to antibody glycosylation, placental inflammation, and impaired trophoblast function. The disruption of transplacental antibody transfer by various pathogens may enhance neonatal morbidity and mortality, necessitating a more complete understanding of the biological and infectious factors impairing antibody transfer. Vaccination during pregnancy has proven to be an increasingly important strategy for neonatal protection from vaccine-preventable diseases during a vulnerable period of life when the immune system is still developing. Fetal and neonatal benefits from maternal vaccination are broad based and greater than the production and transfer of neutralizing antibodies. Disease prevention through vaccination also prevents perturbations in the maternal and neonatal immune response identified in several diseases. By continuing to explore the mechanisms of disease pathogenesis and host responses in pregnant people and neonates, we may optimize the maternal immunological response to immunization and maternal–fetal transfer of antibodies to the newborn.

## Figures and Tables

**Figure 1 vaccines-12-01199-f001:**
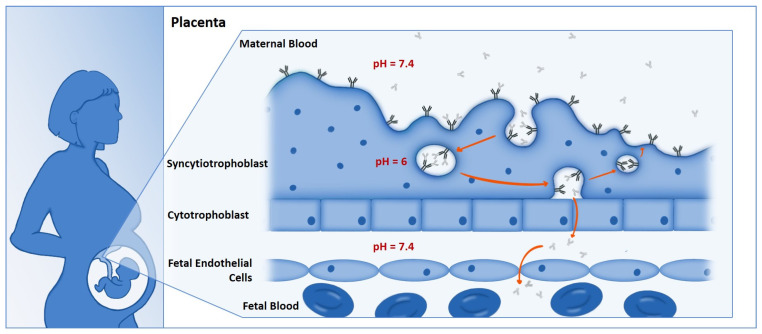
Features of maternal–fetal antibody transfer. Transplacental crossing of IgG and FcRn. Maternal IgG is transferred by transcytosis at the maternal–placental interface at the level of syncytiotrophoblast cells. In the syncytiotrophoblast, endosomes engulf IgG, where it binds to FcRn. The complex travels to the basal cell membrane of syncytiotrophoblast cells and is released into fetal circulation.

**Figure 2 vaccines-12-01199-f002:**
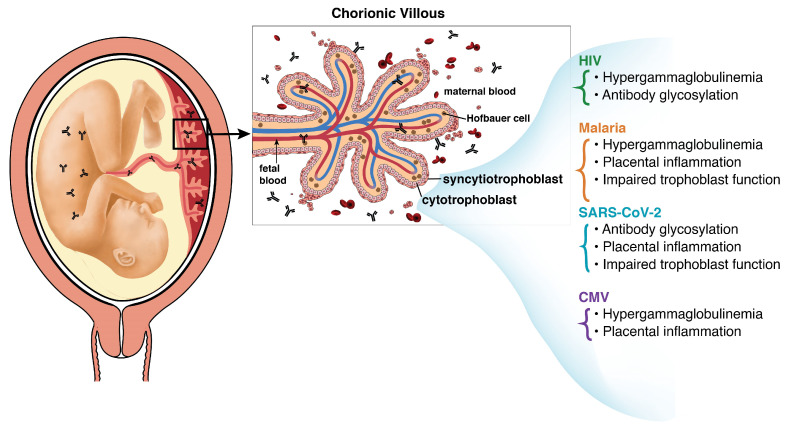
Infection-associated mechanisms that impair transplacental antibody transfer. The mechanisms by which HIV, malaria, SARS-CoV-2, and CMV impair the transfer of maternal antibodies include hypergammaglobulinemia, antibody glycosylation, placental inflammation, and impaired trophoblast function.

**Table 1 vaccines-12-01199-t001:** Current maternal vaccine recommendations in the United States.

Vaccine	Type	Indication
COVID-19	mRNA	Recommended
Tdap	Inactivated	Recommended
Influenza	Inactivated	Recommended
Influenza	Live attenuated	Contraindicated in pregnancy
HPV	Inactivated	Not recommended
Hepatitis A	Inactivated	May be used if indicated (prior to travel, history of injection of illicit drug, professional exposure, chronic liver disease)
Hepatitis B	Protein	May be used if indicated (prior to travel, sexual exposure, intravenous drug use)
Meningococcal	Inactivated	May be used if indicated
MMR	Live attenuated	Contraindicated in pregnancy
PCV 13	Conjugate	No recommendation
Pneumococcal PPSV23	Polysaccharide	Inadequate data
Poliomyelitis	Inactivated	May be used if indicated
Varicella	Live attenuated	Contraindicated
Zoster	Live attenuated	Contraindicated

This table lists the vaccines recommended by the Advisory Committee on Immunization Practices (ACIP). The ACIP develops recommendations for U.S. immunizations that typically become official U.S. Centers for Disease Control policy. Abbreviations: HPV, human papillomavirus; MMR, measles, mumps, rubella; PCV13, pneumococcal conjugate vaccine; PPSV23, pneumococcal conjugate vaccine.

**Table 2 vaccines-12-01199-t002:** Studies on the IgG subclass in uninfected pregnant people.

Subclass Transfer Efficiency	Sample Size	Source
IgG1 > IgG4 > IgG3 > IgG2	N = 69	[37]
IgG1 > IgG3 > IgG4 > IgG2	N = 119	[38]
IgG1 > IgG4 > IgG3 > IgG2	N = 78	[39]
IgG1 > IgG4 > IgG3 > IgG2	N = 27	[40]
IgG3 > IgG2 > IgG1 > IgG4	N = 43	[41]
IgG1 > IgG3 > IgG4 > IgG2	N = 228	[42]
IgG1 > IgG4 > IgG3 > IgG2	N = 107	[31]
IgG1 > IgG4 > IgG3 > IgG2	N = 213	[43]
IgG1 > IgG4 > IgG3 > IgG2	N = 180	[29]
IgG4 > IgG1 > IgG2 > IgG3	N = 34	[44]

This table provides the main conclusions regarding the transfer efficiency of maternal IgG subclasses into the fetus of ten studies that measured paired maternal and cord levels of the four subclasses and had sample sizes > 25 maternal–fetal dyads. Studies utilized radioimmunoassay, immunodiffusion, laser nephelometry, or ELISA to determine transfer efficiency.

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
