# Peer review of "Impact of Infections During Pregnancy on Transplacental Antibody Transfer"

_vaccines, 2024, doi:10.3390/vaccines12101199_

Round 1
Reviewer 1 Report
Comments and Suggestions for Authors
It is a well-described review of transplacental immunoglobulin transfer of antibodies against infectious diseases. The paper summarizes well the indications for various vaccinations during pregnancy. The description of normal placental IgG transfer is correct, but the introduction of pathological conditions focuses only on infectious diseases and their consequences in transplacental antibody transfer. It would be important to also discuss the transfer of natural and pathological autoantibodies. The figures and the table of the paper is informative, the literature cited is adequate.
Author Response
We thank Reviewer 1 for their positive comments.
Suggestion 1: It would be important to also discuss the transfer of natural and pathological autoantibodies.
Author’s response: This is an excellent point. We have now added two paragraphs to discuss the transfer of natural and pathological antibodies and their implication for fetal and neonatal health. The text of these paragraphs is below:
Last paragraph of section 2:
The same mechanism that transfers protective IgG across the placenta also allows natural and pathological maternal antibodies to enter the fetal circulation. Maternal autoantibodies can also transfer to the fetus and may increase the odds for the child to develop autism spectrum disorder, brain injury, or an autoimmune disease later in life (PMID: 17332892, 31191521, 19079257). In Rh alloimmunization, the fetus can be killed by maternal antibodies that target paternally inherited minor blood group antigens (PMID: 29470342). Alternatively, the fetus and neonate may manifest disease (i.e., congenital heart block with lupus) that typically resolves over several months as the maternal antibody concentrations decline (PMID: 22973504). Thus, transplacental antibody transfer has implications for the neonate that extends beyond the provision of anti-infectious protection.
Reviewer 2 Report
Comments and Suggestions for Authors
The manuscript "Impact of Infections During Pregnancy on Transplacental 2 Antibody Transfer" by Celeste Coler et al., reviews the current knowledge of transplacental antibody transfer with the effect of infections. The authors describe the mechanism contributing to the reduction of antibody transfer during maternal infections and identify the knowledge gap which might be important for neonate protection during future pandemics.
Here are my minor suggestions:
Table 2: Please include the target of the IgG subclass e.g. the pathogen. is there any pattern between the IgG subclass transfer efficiency ranking and the class of infected pathogen?
line 330-331: "...showed increased expression of the Fc-330 y receptor III (FCyRII)." III or II?
4.4 Placental inflammation and impaired trophoblast function: Do we have any details about the role of pregnant women's systemic inflammatory levels on placental inflammatory response?
Author Response
Suggestion 1: Table 2: Please include the target of the IgG subclass e.g. the pathogen.
Is there any pattern between the IgG subclass transfer efficiency ranking and the class of infected pathogen?
Author’s response: We have now clarified that Table 2 shows the reports of IgG transfer efficiencies from uninfected pregnant people.
Suggestion 2: line 330-331: "...showed increased expression of the Fc-330 y receptor III (FCyRII)." III or II?
Author’s response: Thank you for noticing this error on our part. The manuscript now reads “(FCyRIII)”.
Suggestion 3: 4.4 Placental inflammation and impaired trophoblast function: Do we have any details about the role of pregnant women's systemic inflammatory levels on placental inflammatory response?
Author’s response: This is an interesting question. It would require an analysis of a matched immunologic response in the mother and placenta, which could only be done if serial biopsies were taken in a study conducted in an animal model with progressively longer intervals between pathogen inoculation and delivery in multiple animals. To our knowledge, this question has not been answered.
Reviewer 3 Report
Comments and Suggestions for Authors
Coler et al describe various aspects of antibody transfer from mother to the fetus and in some cases beyond.
The antibody transfer of different immunoglobulin isotypes via the FcRn is well described and the preference for IgG isotypes well described. A short note as to how long the serum half-lives of maternal antibodies is would be helpful. This is known. It may be further mentioned that the FcRn is not very effective in the gut if at all when IgG is suckled by the baby after birth. IgA is the main immunoglobulin isotype protective. Hence, IgG not taken up during the pregnancy cannot be substituted by breast feeding.
This point should be made clear.
However, important molecules are provided to the baby in maternal milk but may not be the topic of this review but should be mentioned.
The section of infection – associated mechanism impairing maternal – fetal antibody transfer is very important, as the immune reaction of female primates changes drastically due to the placenta. “Manipulations” of the immune system by malaria or HIV alters the immune IgG response against various infective agents not associated with i.e. HIV. This is the reason why ART helps the mother and the baby but maternal antibodies against HIV may not benefit the baby. Similarly, vaccination against COVID 19 benefits the health state of the mother as indicated by various articles. So far, there is fortunately no clear evidence that COVID 19 can induce harmful infections in babies. Hence, maternal vaccinations or natural infection may be addressed with more care explaining what exactly happens.
Monkeypox (Mpox) is an emerging disease in Africa and impacts babies and children in some areas. The expert panel might reflect on this disease as vaccination of mothers against smallpox (variola) a close relative to Mpox clearly benefitted the baby.
Author Response
Suggestion 1: The antibody transfer of different immunoglobulin isotypes via the FcRn is well described and the preference for IgG isotypes well described. A short note as to how long the serum half-lives of maternal antibodies is would be helpful. This is known.
Author’s response: This is an excellent suggestion. We have added a sentence about antibody half-life by IgG isotype in non-pregnant individuals in section 2.
Suggestion 2: It may be further mentioned that the FcRn is not very effective in the gut if at all when IgG is suckled by the baby after birth. IgA is the main immunoglobulin isotype protective. Hence, IgG not taken up during the pregnancy cannot be substituted by breast feeding. However, important molecules are provided to the baby in maternal milk but may not be the topic of this review but should be mentioned.
Author’s response: Thank you for making this point. It is important that we state why deficiencies in transplacental antibody transfer cannot be entirely compensated for through breast feeding. Some studies argue that the reduced transfer of IgG in breast milk is primarily because human colostrum has a low IgG content. Others argue that reduced transfer is due to gut closure, a developmental stage of intestinal maturation during which neonatal intestinal permeability to macromolecules and IgG decreases. Many of the studies showing decreased efficiency of IgG transfer through breast milk have been performed in rodent or bovine models, in which the gut epithelium differs in composition from humans. Additionally, placentally transferred IgG provides systemic immunity, whereas antibodies from breast milk largely provide neonatal mucosal immunity. In the manuscript, we have now added language to state that IgG is not transferred to the neonate in meaningful amounts in breast milk, where IgA predominates.
Suggestion 3: “Manipulations” of the immune system by malaria or HIV alters the immune IgG response against various infective agents not associated with i.e. HIV. This is the reason why ART helps the mother and the baby but maternal antibodies against HIV may not benefit the baby. Similarly, vaccination against COVID 19 benefits the health state of the mother as indicated by various articles. So far, there is fortunately no clear evidence that COVID 19 can induce harmful infections in babies. Hence, maternal vaccinations or natural infection may be addressed with more care explaining what exactly happens.
Author’s response: We agree that the benefits of maternal vaccination against disease are broader than simply producing neutralizing antibodies, because protective immunity prevents broad perturbations in the maternal immune response. We have now included a sentence in order to make that point.
Suggestion 4: Monkeypox (Mpox) is an emerging disease in Africa and impacts babies and children in some areas. The expert panel might reflect on this disease as vaccination of mothers against smallpox (variola) a close relative to Mpox clearly benefitted the baby.
Author’s response: We recently wrote a review article on this topic and are happy to mention MPOX and the benefit of vaccinating against smallpox. This is another excellent example of a broader health benefit from vaccination. We have included a sentence about how expanding our knowledge of transplacental antibody transfer and pathogen induced damage to the placenta can inform prophylactics and treatments against emerging pathogens.
Round 2
Reviewer 3 Report
Comments and Suggestions for Authors
Good work
Author Response
Thank you for your positive comments on our manuscript. I don't see any specific recommendations for revision, but please let us know if there is something you would like us to revise.